SEMGROMI—a semantic grouping algorithm to identifying microservices using semantic similarity of user stories

http://orcid.org/0000-0003-4003-497X Vera-Rivera Fredy H. 1 fredyhumbertovera@ufps.edu.co
Puerto Cuadros Eduard Gilberto 1
http://orcid.org/0000-0001-9249-1756 Perez Boris 1
http://orcid.org/0000-0002-6487-5813 Astudillo Hernán 2 4
Gaona Carlos 3
1 Grupo de Investigación GIA, Universidad Francisco de Paula Santander , Cúcuta, Norte de Santander , Colombia
2 Departamento de Informática, Universidad Técnica Federico Santa María , Santiago, Santiago , Chile
3 Grupo de Investigación GEDI, Universidad del Valle , Cali, Valle del Cauca , Colombia
4 Instituto de Tecnología para la Innovación en Salud y Bienestar, Universidad Andrés Bello , Viña del Mar , Chile
Ardito Luca
Electronic publication date: 2023 May 12
Publication date: 2023
Volume: 9
Electronic Location ID: e1380
Received 2022 Nov 16; Accepted 2023 Apr 13
Copyright: © 2023 Vera-Rivera et al.
Copyright year: 2023
Copyright holder: Vera-Rivera et al.
License: This is an open access article distributed under the terms of the Creative Commons Attribution License, which permits unrestricted use, distribution, reproduction and adaptation in any medium and for any purpose provided that it is properly attributed. For attribution, the original author(s), title, publication source (PeerJ Computer Science) and either DOI or URL of the article must be cited.
License URL: https://creativecommons.org/licenses/by/4.0/

Keywords: Microservices, Micro-services granularity, Semantic similarity, User stories, Services computing, Micro-services decompositions

Funding: Colombia’s Ministry of Science and Technology (Minciencias-Colciencias) 753-Formación de capital humano de alto nivel para el departamento Norte de Santander Francisco de Paula Santander University (Cúcuta, Colombia) 14 of 2016 Universidad del Valle (Cali, Colombia) ANID (Chile) through Anillo ACT210021 This work was supported by Colombia’s Ministry of Science and Technology (Minciencias-Colciencias) through doctoral scholarship “753-Formación de capital humano de alto nivel para el departamento Norte de Santander”; by the Francisco de Paula Santander University (Cúcuta, Colombia) through the doctoral studies commission number 14 of 2016; by the Universidad del Valle (Cali, Colombia); and by ANID (Chile) through Anillo ACT210021 Aconcagua. There was no additional external funding received for this study. The funders had no role in study design, data collection and analysis, decision to publish, or preparation of the manuscript.

==============================
Microservices is an architectural style for service-oriented distributed computing, and is being widely adopted in several domains, including autonomous vehicles, sensor networks, IoT systems, energy systems, telecommunications networks and telemedicine systems. When migrating a monolithic system to a microservices architecture, one of the key design problems is the “microservice granularity definition”, i.e., deciding how many microservices are needed and allocating computations among them. This article describes a semantic grouping algorithm (SEMGROMI), a technique that takes user stories, a well-known functional requirements specification technique, and identifies number and scope of candidate microservices using semantic similarity of the user stories’ textual description, while optimizing for low coupling, high cohesion, and high semantic similarity. Using the technique in four validation projects (two state-of-the-art projects and two industry projects), the proposed technique was compared with domain-driven design (DDD), the most frequent method used to identify microservices, and with a genetic algorithm previously proposed as part of the Microservices Backlog model. We found that SEMGROMI yields decompositions of user stories to microservices with high cohesion (from the semantic point of view) and low coupling, the complexity was reduced, also the communication between microservices and the estimated development time was decreased. Therefore, SEMGROMI is a viable option for the design and evaluation of microservices-based applications. The proposed semantic similarity-based technique (SEMGROMI) is part of the Microservices Backlog model, which allows to evaluate candidate microservices graphically and based on metrics to make design-time decisions about the architecture of the microservices-based application.

Introduction

The development of software systems for modern technologies has seen exponential growth in the last decade (Tanveer, 2015). The demands for information processing continue to grow, requiring specialized software with strong capabilities in terms of security, reliability, robustness and interoperability with other systems, especially with mobile applications and with a smart world that is evolving every day. Services enable communication between heterogeneous software systems, devices and applications. Enterprise transaction processing is accomplished by connecting disparate software systems through business services. Microservices architecture allows large applications to be implemented and deployed as a collection of cloud-run services, and is a topic of great interest in both academia and industry (Newman, 2015).

A key concept in microservices design is granularity (Vera-Rivera et al., 2021), which describes all the microservices that make up the system and the size (number of services or operations) of each of them. Hassan, Bahsoon & Kazman (2020) pointed out that granularity is determined by both the size of the service and the amount of functionality it exposes. Not surprisingly, defining the “right” level of granularity for a microservice is an active research topic.

Granularity can be achieved and then improved by joining or splitting microservices. The problem of microservice granularity is presented as a problem of boundary (size) identification for the microservice itself (Homay et al., 2020). The microservice requires two properties to be defined: (i) a specific and decoupled purpose, and (ii) added value to the system. Granularity can be beneficial if it increases the modularity of the system (flexibility, scalability, maintainability, and traceability) while reducing complexity (dependency, communication, and computation).

In this article, we address the granularity problem by proposing a semantic grouping algorithm called SEMGROMI. This proposal focuses on grouping user stories into microservices considering cohesion (high), coupling (low), and complexity (low) among the identified microservices. The technique presented in this article is semi-automatic, where the software architect identifies and analyzes the resulting solution and makes design-time decisions about that solution based on metrics. SEMGROMI is part of the Microservices Backlog model (Vera-Rivera et al., 2021).

The next section of the article introduces the Microservices Backlog (MB) model. An overview of related work on methods or techniques for defining microservices granularity is presented in “Related Work”. The methodology and evaluation methods are described in “Methodology”. After that, “SEMGROMI: Semantic Grouping Algorithm” presents the semantic similarity grouping algorithm, SEMGROMI, in detail “Validation” describes the results of the evaluation methods. Finally, we discuss the results of the technique and summarize and conclude our discussion in “Discussion” and “Conclusions”.

Microservices backlog

In Vera-Rivera et al. (2021), we proposed the Microservices Backlog (MB), a method for graphically analyzing microservices granularity at design time; allowing architects to analyze and make build decisions about the application and its dependencies. MB answers three key questions: (1) How do you determine and evaluate the granularity of microservices? (2) How to determine the number of user stories assigned to each microservice, and (3) How to determine the optimal number of microservices that will be part of the application? These activities make it possible to improve the granularity of the microservices to achieve low coupling, high cohesion, and low complexity properties. These design-time metrics have been adapted and computed to evaluate the granularity of a microservice-based application (Vera-Rivera et al., 2021). The problem of assigning user stories to microservices has high complexity, increasing the number of user stories significantly increases the execution time of the genetic algorithm.

This article describes SEMGROMI, a semantic similarity-based grouping technique that overcomes the runtime limitations of MB, allowing the granularity of microservices to be defined with a drastic reduction in execution time, while achieving results with similar coupling, complexity, and cohesion. We validated SEMGROMI with the same three projects used to validate MB, namely Cargo Tracking (Baresi, Garriga & De Renzis, 2017; Li et al., 2019), JPetStore (Jin et al., 2019; Saidani et al., 2019, Ren et al., 2018) and Foristom Conferences (Vera-Rivera et al., 2021). We have also evaluated it with Sinplafut (Vera-Rivera, Vera-Rivera & Gaona-Cuevas, 2019), an industrial case study with 92 user stories. Design time metrics were adapted and calculated to evaluate the granularity level of each microservice in the proposed decomposition.

The main contributions of this work are: (i) a semantic similarity-based grouping algorithm for aggregating user stories into microservices, and (ii) design-time metrics adapted and used for both identifying microservices and evaluating the technique itself. (iii) an improved runtime for decomposing user stories into microservices compared to a genetic algorithm.

The Microservice Backlog (MB), from a set of functional requirements expressed as user stories within a product backlog or release planning, allows the granularity of microservices to be analyzed graphically. MB provides suggested architectures for microservices-based applications, allowing architects or development teams to evaluate the granularity or size of microservices, considering at design time their complexity, coupling, cohesion, calls and requests between microservices, and estimated development time. This allows architects and developers to find an implementation strategy (Vera-Rivera et al., 2021).

The architect creates the project and submits the user story data (i.e., identifier, name, description, estimated points, estimated development time, scenario, and observations) from a CSV file. The user then defines dependencies between user stories (HU) according to the business logic. A trace between HUi and HUj is defined when HUi calls or executes HUj. Users and architects can add up these user stories and generate an automatic decomposition into microservices (using a genetic algorithm or the SEMGROMI grouping algorithm), or define the decomposition manually. The system calculates the metrics for analyzing the application using the metrics calculator component. With these metrics, you can analyze the proposed architectures of the project at design time and make decisions Vera-Rivera et al. (2021).

The genetic algorithm considers coupling, complexity, semantic similarity and cohesion metrics to automatically distribute user stories to microservices. It tries to find the best combination and assignment of user stories to microservices, thus minimizing the granularity metric (Vera-Rivera et al., 2020). It is important to note that semantics and conceptual similarity play a big role in several topics around architecture, including architecture recovery, refactoring (e.g., Feature Envy, Move Method, Extract Class), and design principles (single responsibility principle). For this reason, the grouping algorithm SEMGROMI considers semantic similarity to identify the topic to which the user story refers and to group into microservices those that refer to the same topic. The details of the algorithm are presented later in the article.

Related work

According to Vera-Rivera, Gaona & Astudillo (2021), the problem of microservice identification has been approached from several perspectives, including clustering in machine learning, domain engineering, genetic programming and semantic similarity as the most studied. The granularity of microservices is evaluated using metrics, in particular, to measure performance and to measure the degree of coupling. These two metrics are the most widely used for this purpose. Table 1 shows a comparative analysis of related work.

Table 1 Related works microservices granularity definition (Vera-Rivera et al., 2021).

Year	Articles	Metrics	Quality Att.	Technique, method	
2022	Semgromi	Complexity, coupling, cohesion, granularity, performance: microservices calls.	Modularity, maintainability, functionality, performance.	Hierarchical clustering, semantic similarity (Natural processing language),	
2021	Genetic programming	Complexity, coupling, cohesion, granularity, performance: microservices calls.	Modularity, maintainability, functionality, performance.	Genetic programming, semantic similarity (Natural processing language)	
2020	2	Cohesion, granularity	None	Domain-driven design, architectural design via dynamic software visualization. Clustering using affinity propagation algorithm, and clustering of semantically similar.	
2019	12	Coupling, cohesion, granularity, computational resource, performance, source code.	Scalability, performance, functionality, modularity, maintainability.	Machine learning: K-means, dataflow driven descomposition, DISCO, non-dominated sorting genetic algorithm, hierarchical clustering, semantic similarity.	
2018	6	Coupling, cohesion, complexity, granularity, computational resource, performance	Scalability, performance, availability.	Domain engineering, domain-driven design, domain-driven design COSMIC function points, functional decomposition, heuristics used for functional splitting, microservice discovery algorithms, decomposition pattern.	
2017	7	Performance	Scalability, performance, reliability, maintainability	Vertical decomposition, balance cost quality assurance vs deployment, architecture definition language (ADL), semantic similarity, clustering k-means, DISCO, graph-based clustering algorithm, virtual machine image synthesis and analysis	
2016	2	Coupling, security, and scalability impact.	Scalability, security	Self-adaptative solution. Decomposition from system requirements—security vs scalability.	

Machine learning clustering includes clustering K-means (Baresi, Garriga & De Renzis, 2017; Ren et al., 2018), scale-weighted K-means (Abdullah, Iqbal & Erradi, 2019), graph-based clustering (Mazlami, Cito & Leitner, 2017), hierarchical clustering (Al-Debagy & Martinek, 2019; Nunes, Santos & Rito Silva, 2019) and affinity propagation (Al-Debagy & Martinek, 2020).

Other techniques also used to identify microservices are domain engineering and domain-driven design (DDD) (Josélyne et al., 2018; Krause et al., 2020), COSMIC function points (Vural, Koyuncu & Misra, 2018), functional decomposition (Tyszberowicz et al., 2018; Baresi, Garriga & De Renzis, 2017), class-based extraction model (Mazlami, Cito & Leitner, 2017), data flow-driven decomposition algorithm (Chen, Li & Li, 2017), functional partitioning through heuristics and microservices discovery algorithms (De Alwis et al., 2018), process mining (Taibi & Syst, 2019), and service cutter, a method for decomposing a service (Gysel et al., 2016).

In terms of genetic programming techniques to address microservice granularity, the most commonly used are: (i) NSGAII, a non-dominated sorting genetic algorithm II—(Jin et al., 2019; De Alwis et al., 2019; Saidani et al., 2019) and (ii) multi-objective genetic algorithm (Christoforou, Odysseos & Andreou, 2019).

Semantic similarity has also been used (Baresi, Garriga & De Renzis, 2017; Taibi & Syst, 2019; Al-Debagy & Martinek, 2019); intuitively, high semantic similarity correlates with high cohesion because it groups together services related to the same concept or domain (Perepletchikov, Ryan & Frampton, 2007; Candela et al., 2016).

Baresi, Garriga & De Renzis (2017) proposed a lightweight semantic analysis process to support the identification of candidate microservices. The solution is based on the semantic similarity of the predicted/available functionality described by the OpenAPI specifications. The process is supported by a fitness function to perform a mapping of the available OpenAPI specifications to the entries of a reference vocabulary.

Taibi & Syst (2019) proposed a three-step mining approach to identify business processes in monolithic solutions: (i) The DISCO (extracting distributionally related words using co-occurrences) tool is used to identify business processes. (ii) A set of microservices is proposed based on business processes with similar behavior. This behavior is related to the execution paths of the processes. Care is taken to avoid circular dependencies. And (iii) the quality of the decomposition is evaluated using a set of metrics proposed by them.

Al-Debagy & Martinek (2020) propose a method consisting of several steps, starting with the operation names extracted from the OpenAPI specifications. The second step is the process of converting the operation names into word representation using word embedding models. The third step is the clustering of semantically similar operation names to create candidate microservices.

These methods are mainly used for scenarios where monolithic architectures are migrating to microservices architectures and focus mainly on the design phase. Cohesion, coupling and performance metrics were the most commonly used metrics to assess granularity; most techniques are manual, with only a few automated or semi-automated.

Few methods support the development of microservices architectures from scratch (greenfield development). Source code, call graphs, logs and use case descriptions are the most commonly used inputs for this development. However, none of the surveyed work has used user stories as input to define microservices. User stories define microservice operations or services as requirements. Finally, none of the surveyed studies use data from agile practices or agile software development (Vera-Rivera, Gaona & Astudillo, 2021).

For a more detailed comparative analysis of the related works please see Vera-Rivera, Gaona & Astudillo (2021).

Methodology

Design science research (Hevner et al., 2004) was used to conduct this research. This research paradigm aims to improve the creation of innovative artifacts through a continuous and iterative process, in this case the grouping algorithm itself. Our evaluation process consists of six research activities. The adaptation of this paradigm to our problem is shown in Fig. 1. Identify and frame the problem: Review the current state of studies related to ours to identify gaps, and identify metrics to evaluate the granularity of microservices (Vera-Rivera, Gaona Cuevas & Astudillo, 2019).

Design grouping algorithm: Once the metrics have been identified, the grouping algorithm is designed. The algorithm has six parts: (1) semantic similarity calculator, (2) semantic grouper, (3) call and request calculator, (4) dependency analyser, (5) coupling grouper, and (6) metric calculator.

Create grouping algorithm: It was implemented in Python using the artificial intelligence and natural language processing libraries Spacy (Spacy.io, 2020) for vector algebra and distance between points.

Evaluation of a case study: The evaluation was carried out in two academic projects (Cargo Tracking (Baresi, Garriga & De Renzis, 2017) and JPetStore (Jin et al., 2019)) and two industry projects (Foristom Conferences and Sinplafut (Vera-Rivera, Vera-Rivera & Gaona-Cuevas, 2019)).

Compare the results of the algorithm with other methods: We compared the decompositions proposed by our algorithm with those proposed by other state-of-the-art microservices identification methods: domain-driven design (DDD), service cutter (Gysel et al., 2016), Microservices identification trough interface analysis (MITIA) (Baresi, Garriga & De Renzis, 2017), and our own genetic programming technique (Vera-Rivera, Gaona & Astudillo, 2021). We compared these approaches using coupling, cohesion, complexity, granularity, development time and performance metrics.

Proposal for an algorithm to group user stories into microservices: After carrying out evaluations and appropriate adjustments, the algorithm for grouping user stories into microservices is proposed in this article.

Figure 1 Research model.

Semgromi: semantic grouping algorithm

The problem is to distribute k microservices n user stories, grouping stories with the highest semantic similarity, i.e., grouping stories that refer to the same topic. The grouped stories should also have low coupling and high cohesion. There is no fixed number of microservices. It is not convenient to determine in advance how many microservices the application must have.

To solve this problem, and taking into account the above considerations, the grouping algorithm (see Fig. 2) is a semi-automatic approach where the user or architect can iteratively analyze the proposed solution and run it repeatedly until the goals of low coupling, high cohesion and high semantic similarity are achieved. The grouping algorithm has three parts: (1) defining parameters, (2) grouping user stories by semantic similarity, and (3) grouping microservices by semantic similarity.

Figure 2 Grouping algorithm design.

Algorithm parameters

The grouping algorithm has a number of input parameters: Semantic similarity threshold: Since the semantic similarity between two texts is a value between 0 and 1 (with one being the same or very similar), this is the minimum acceptable similarity value above which stories will be grouped; its initial default value is 0.85. This value indicates the percentage of similarity that exists between user stories; they are considered to be similar if the semantic similarity value is greater than 85%. This value can be adjusted by the user or the architect. The semantic similarity threshold indicates how similar the texts must be to be grouped, a value closer to 1 indicates that the semantic similarity must be higher to be grouped into the same microservice; therefore, more microservices are obtained. Smaller values will group more stories into fewer microservices.

Coupling threshold: the minimum acceptable value of the degree of coupling between the stories to be grouped; its initial default value is 0.5. The microservices whose distance is greater than the coupling threshold are grouped together. If the value is close to 1.0, more microservices will be obtained because few will exceed that value. On the other hand, if its value is close to 0.0, only one microservice will be obtained, because all of them exceed that value. This parameter can be changed by the user, initially and according to the tests performed its value is 0.5.

Language: the natural language in which the user stories are written; currently only Spanish and English are supported.

Use entity lemmatization vs full text: whether to use the full text of the user stories or only entity lemmas to calculate semantic similarity.

Group user stories by semantic similarity

User stories are grouped into candidate microservices according to their semantic similarity, in three steps as follows.

A user story describes the functionality that will be part of the software system, it will be value to a user or customer Cohn (2004, 2005). We defined a template for the user stories definition, according to Beck (2000), as follows: User story identifier, Name, Description, Sprint, Date, Priority, Estimated points, Estimated development time, End date, Actor or role, Developer, Additional details (photo, image, document, video), Task list, Restrictions, Acceptance criteria, Definition of done, and Dependencies.

This information is used to define the dependencies between the user stories, to calculate the evaluation metrics, and define the SEMGROMI algorithm.

Semantic similarity among user stories

User stories are written in prose and define the functional requirement that the application must implement; their description details the users who will use the functional requirement and the actions that the system must perform on the application’s business logic. The name and description of all user stories are merged into a single text; verbs, articles, adjectives and prepositions are removed from the text, leaving only nouns, because nouns correspond to the entities involved in the user story. The entities are the objects on which the action to be performed by the application is performed. The topics of the text were identified by counting the number of times an entity is repeated. The most repeated words were selected. These words are stored in a string for semantic similarity calculations. The similarity between user stories is calculated using the frequency of each domain entity in each user story, resulting in a semantic similarity dictionary between user stories as shown in Eq. (1):

(1) DSS={<“hu1−hu2”,a1−2>,…,<“huj−huk”,aj−k>}

where: DSS: the semantic similarity dictionary among user stories.

huj and huk: the user stories’ identifier; it is used as dictionary key, which is formed by concatenating the identifiers of the user stories that are compared.

aj−k are the dictionary values, it is the semantic similarity value between the huj and huk user stories (obtained by Spacy), which is in the 0..1 interval: the closer to 1, the more similar the user stories are.

Semantic similarity among user stories was implemented with the Spacy natural language processing library (Spacy.io, 2020), which uses artificial neural networks and was designed to be used in production.

For example, given the following user stories, their semantic similarity is calculated in the following way:

Id: Hu1

Name: Create voyage of Cargo.

Description: As a user I need to create a Voyage for a given cargo, specifying the locations needed to reach its destination.

Id: Hu2

Name: Handle Cargo Event

Description: As a user I need to create a cargo event in a location indicating the date when the event ends.

Id: Hu3

Name: Get locations

Description: As a user I need to obtain a list of available locations.

Now we show how the semantic similarity between user stories is calculated:

1. Join name and description.

Text1: Create voyage of Cargo. As a user I need to create a Voyage for a given cargo, specifying the locations needed to reach its destination.

Text2: Handle Cargo Event. As a user I need to create a cargo event in a location indicating the date when the event ends.

Text3: Get locations. As a user I need to obtain a list of available locations.

2. We removed verbs, articles, adjectives and prepositions from the text, leaving only nouns.

Text1: voyage cargo. user voyage cargo, locations destination.

Text2: Cargo Event. user cargo event location date event.

Text3: locations. user list locations.

3. The topics of the text were identified by counting the number of times an entity is repeated.

Text1: Voyage(2), cargo(2), user(1), locations(1), destination(1) then Text1: Voyage cargo user

Text2: Cargo(2), event(3), user(1), location(1), date(1) then Text2: Event cargo user

Text3: Locations(2), user(1), list(1) then Text3: Locations user list

4. Calculate the semantic similarity.

DSS=[<“Hu1−Hu2”,0.87>,<“Hu1−H3”,0.65>,<“Hu2−Hu3”,0.55>]

With the semantic similarity dictionary we proceed with the grouping process.

Grouping user stories

The grouping process is similar to hierarchical clustering (Han, Kamber & Jian, 2012) with some variations; from the list of user stories, the similarity between each pair of user stories HUi and HUj is computed, and if it exceeds the semantic similarity threshold, they are included in the same microservice, otherwise they are left in different microservices. In this iterative process, each user story is compared to each candidate microservice and grouped where its semantic similarity is highest; if the story is not similar to any of the candidate microservices, a new microservice containing that story is added.

The semantic similarity of each microservice is computed using the assignment of user stories to each microservice and the semantic similarity dictionary (see Eq. (2)), and the semantic similarity of the entire application (Eq. (3)).

(2) SSi=1/c∑j=1,k=j+1maj−k

where: 1. SSi: semantic similarity of the i−th microservice.

2. c: number of comparisons made to calculate the semantic similarity of the microservice’s user stories. It is used to calculate the average semantic similarity of the microservice. For example, for a microservice that has two user stories assigned to it (hu1,hu2), the semantic similarity between hu1 and hu2 is calculated once, a single comparison (c=1). If the microservice has three user stories, it must compare (hu1,hu2), (hu1,hu3), and (hu2,hu3), corresponding to three comparisons (c=3).

3. m: number of microservice’s user stories.

4. aj−k corresponds to the dictionary value, they are the semantic similarity values between the huj and huk.

(3) SsT=100/n∑i=1nSSi

where: 1. SsT: the total semantic similarity of the application, it was the average of the semantic similarity of each microservice. To obtain a semantic similarity value between 0 and 100, we multiply the average by 100.

2. SSi: semantic similarity of the i−th microservice.

3. n: number of microservices of the application.

Group microsevices using interdependence

For each pair MSi and MSj, the algorithm computes the number of times MSi calls MSj (callsi) and vice versa, (requesti). If both values are greater than zero, then microservices MSi and MSj are said to be interdependent and must therefore be merged into one. This process is performed iteratively for each microservice; since the resulting microservices may have interdependence, the designer can repeat the process (if desired). if we adopt such rule as it is, the system will end up with a single microservice.

In some real cases the interdependence between microservices must be accepted (a threshold value), in each iteration the user evaluates the solution obtained and decides whether to run the algorithm again or to accept that solution.

Group microservices by coupling distance

The microservices whose distance is greater than the coupling grouping threshold will be grouped (remember that the initial default value of this parameter is 0.5, but it can be modified). It tries to reduce the high communication or calls between the microservices of the application; if two microservices have many dependencies, they should be grouped together, thus reducing the coupling. If two microservices have many calls, their dependency is high; if one microservice changes, it is possible that the other will have to change, so they have high coupling. The goal is to reduce that high dependency.

Computing the coupling between microservices

For each pair of microservices, the algorithm calculates their coupling CpD, based on the calls and requests between them, see Eq. (4).

(4) CpDi−j=(callsi−j+requesti−j)/total_calls

where: CpDi−j: coupling distance between microservices i and j.

callsi−j: number of times microservice i calls microservice j (i.e., i’s inputs).

requesti−j: number of times microservice j calls microservice i (i.e., i, outputs).

total_calls: the total number of calls among the application microservices.

Group microservices by semantic similarity

After reducing the interdependent microservices and reducing the microservices with the largest coupling, the semantic similarity of the microservices is checked. If any microservices are semantically similar, they are grouped together. This increases the cohesion of the microservices because it groups the microservices that are related to the same topic or theme. This process is described in more detail below: 1. Identify the entities of the microservices: For each microservice, the name and description of all associated user stories are combined into a global text that is lemmatized to identify its domain entities (as denoted by nouns).

2. Computes the semantic similarity between the microservice entities: The frequency of each entity in each microservice is calculated, and the two entities with the highest frequency are selected to perform the semantic comparisons between the microservices. This process uses the machine learning technique of text classification, which automatically assigns tags or categories to text. The result is the semantic similarity between each pair of microservices, which is computed iteratively to create a semantic similarity matrix.

3. Group microservices by semantic similarity: Microservices are grouped using the same algorithm that groups user stories, but now using the semantic similarity between microservices (as defined above). For each pair of microservices, they are joined if their semantic similarity value is above the semantic similarity grouping threshold; its initial and default value is 0.85, but can be changed by the designer.

Calculate metrics and evaluate candidate microservices

The technique calculates the metrics and draws the microservices backlog diagram to evaluate the candidate microservices. The user evaluates the obtained solution, if it does not meet the desired requirements, if it still has high coupling, low cohesion and high complexity, he can repeat the microservices grouping process until he finds an adequate solution.

The resulting full pseudo-code of the algorithm is shown in Algorithm 1.

Algorithm 1 SEMGROMI: semantic grouping algorithm of microservices

    // Process A - Define parameters as input data	
    input: list[UserStories], list[dependences], similarityParameter, couplingParameter, language, semanticOn	
    output: list[microservice, metrics]	
    begin	
      // Process B - Group user stories by semantic similarity	
       listEntities[userStory,text,lemmas]←identifyEntitiesFrequency(list[UserStory]);	
       listMs[microservice]←	
       groupByEntityFrequency(listEntities, semanticOn, similarityParameter);	
      // Process C - Group interdependent microservices	
       matrixCalls←calculateCallsRequest(listMS);	
      if interdependentMS then	
         listMSCandidate←groupbyInterdepentMicroservices(listMS,matrixCalls);	
        else	
          // Process D - Group microservices by coupling distance	
          distanceMatrix←calculateCouplingDistance(listMS,matrixCalls);	
         if distance > couplingParameter then	
            listMSCandidate←groupbyCouplingDistance(listMS,distanceMatrix);	
           else	
             // Process E - Group microservices by semantic similarity	
              matrixMSEntities←identifyEntitiesFrequency(listMS);	
              listMSCandidate←;	
              groupBySemanticSimilarity(listaMS, matrixMSEntities);	
           end	
         end	
        end	
      end	
      // Process F - Calculate metrics	
       metrics←calculateMetrics(listMSCandidate);	
      drawMicroservicesBacklog();	
      return list[ListMSCandidate, metrics]	
    end	

Validation

The proposed technique was validated with actual projects from a state-of-the-art review (Vera-Rivera, Gaona & Astudillo, 2021; Vera-Rivera, Gaona Cuevas & Astudillo, 2019). We did not find a catalog of projects with user stories available for testing and comparing state-of-the-art methods, but we were able to use four interesting candidates: two educational projects (Cargo Tracking (Baresi, Garriga & De Renzis, 2017; Li et al., 2019) and JPetStore (Jin et al., 2019; Saidani et al., 2019; Ren et al., 2018)) and two industry projects (Foristom Conferences (Vera-Rivera et al., 2021) and Sinplafut (Vera-Rivera, Vera-Rivera & Gaona-Cuevas, 2019)). In general, it’s hard to find test data from software systems developed with microservices that also include a requirements document or user stories. Therefore, we had to recover the user stories from the available documentation for the study cases.

Evaluation methodology

The technique was subjected to an observational and analytical evaluation with these cases, following the recommendations of Hevner et al. (2004). The evaluation compared the metrics of the decompositions obtained by other methods described in the literature: Domain-Driven Design (DDD) (Evans, 2015), Service Cutter (Gysel et al., 2016), Microservices Identification through Interface Analysis (MITIA) (Baresi, Garriga & De Renzis, 2017), and Identification of Candidate Services of Monolithic Systems based on Execution Traces (Jin et al., 2019). Since DDD is currently the most widely used method for microservice identification, we used it as a kind of “sanity check” to verify that SEMGROMI’s decomposition was consistent and relatively close.

The analytical evaluation included both static and dynamic measurements. Metrics for coupling, complexity, cohesion, dependencies, performance, and size of the proposed decomposition (or microservices-based application) were compared between the result of our technique and other state-of-the-art approaches. These metrics were computed at design time from extracted user stories and their dependencies; the same user stories, dependencies, and computations were used in all tests. The evaluation process was as follows 1. The state-of-the-art examples and industry projects were analyzed and described.

2. The user stories of each example and project were identified to obtain the “product backlog”.

3. User story dependencies were identified according to data flow, calls, invocations between user stories or business logic.

4. Decompositions were obtained with Microservices Backlog (using the previous genetic algorithm and this new grouping algorithm), and the decompositions obtained with the other methods were uploaded to the system.

5. The metric calculator obtained the metrics and the dependency graph of the Microservices Backlog of the candidate microservices for each decomposition.

6. The metrics for each decomposition were evaluated and compared.

The evaluation data set (projects)

The project details are presented in the Table 2, which summarizes for each project (1) the number of user stories (2) the total number of story points, jointly estimated by the co-authors of this article, and (3) the total estimated development effort (in hours), an indication of the complexity and size of the project.

Table 2 Projects for evaluating the grouping algorithm

Name	User stories	Points	Dev. time (H)	
Cargo tracking	14	51	77	
JPet store	22	73	115	
Foristom conferences	29	235	469	
Sinplafut	92	302	604	

The evaluation metrics

Several metrics have been adapted from Bogner, Wagner & Zimmermann (2017), Rud, Schmietendorf & Dumke (2006), and our own previous work (Vera-Rivera, Gaona & Astudillo, 2021) to compare the decompositions obtained by each method. Microservices Backlog calculates metrics for coupling, cohesion, granularity, complexity, development time, and performance. The metrics used are • Granularity—N: Number of microservices of the decomposition or system.

• Coupling (CpT): the absolute importance of the microservice (AIS), absolute dependence of the microservice (ADS), and microservices interdependence (SIY).

AISi is the number of clients invoking at least one operation of MSi; to calculate the total value of AIS at the system level (AisT) the vector norm is calculated. See Eq. (5).

(5) AisT=|AIS→|=AIS12+AIS22+⋯+AISN2

ADSi is the number of other microservices on which the MSi depends. To calculate the total value of ADS at the system level (AdsT) the ADS→ vector norm is calculated (See Eq. (6)). Then:

(6) AdsT=|ADS→|=ADS12+ADS22+⋯+ADSN2

SIYi defines the number of pairs of microservices that depend bi-directionally on each other divided by the total number of microservices (Eq. (7)).

(7) SiyT=|SIY→|=SIY12+SIY22+⋯+SIYN2

Calculating the norm of the vector Cp→ we have the coupling value for the application (CpT), Eq. (8):

(8) CpT=10∗|Cp→|=SIY12+SIY22+⋯+SIYN2

where Cp→=[AisT,AdsT,SiyT], we amplify CpT by 10, in such a way that its dimension is like the dimension of the other metrics. • Cohesion—Lack of cohesion (CohT): the number of pairs of microservices not having any dependency between them, adapted from Candela et al. (2016). Lack of cohesion of a microservice is the number of pairs of microservices that have no interdependency between them. The LC of MSi has been defined by us as the number of pairs of microservices that have no interdependency between MSi. The lack of cohesion degree (Cohi) of each microservice i is the ratio of LC and the total number of microservices in the application (Eq. (9)), and CohT is the vector norm of the vector consisting of the Coh value of each microservice of the application (Eq. (10)).

(9) Cohi=LCi/N

(10) CohT=|Coh→|=Coh12+Coh22+⋯+CohN2

where Coh→=[Coh1,Coh2,…,CohN] • Cohesion—Total Semantic similarity (SsT): the average of the semantic similarity of each microservice (see Eqs. (2) and (3)).

• Granularity—Weighted Service Interface Count (WSICi): is the number of exposed interface operations of the microservice i (Hirzalla, Cleland-Huang & Arsanjani, 2009). We assume that each user story is associated with a single operation, so we adapt this metric as the number of user stories associated with a microservice, and WsicT is the highest WSICi of the system decomposition (Eq. (11)).

(11) WsicT=Max(WSIC1+WSIC2+⋯+WSICN)

• Performance— Calls: the total number of invocations between microservices.

• Performance— Avg.Calls: average of calls that a microservice makes to another: Calls/N.

• Complexity—Story Points Max.(Pi): estimated effort needed to develop a user story; Max.(Pi) is the largest number of story points associated with any microservice.

• Complexity—Cognitive complexity points (CxT): estimated difficulty of developing and maintaining a microservice-based application, using its estimated story points, relationships, and dependencies among microservices (see more details in Vera-Rivera et al. (2021)).

(12) Cx=(∑1NCgi)+Max(P1,…,PN)+(N∗WsicT)+(∑1NPfi)+(∑1NSIYi)

(13) CxT=Cx/Cx0

where: CxT: Cognitive complexity points of the system.

i: ith microservices.

Cgi: Pi∗(Callsi+Requesti), Callsi are the outputs of MSi and Requesti are the inputs of MSi.

Pi: Total user story points of MSi. Max(P1,…,PN) Maximum Pi of the system.

Pfi: Number of nodes used sequentially from a call that makes a microservice to other microservices, counted from the i−th microservice; A larger depth implies a greater complexity of implementing and maintaining the application.

Cx0: The base case where the application has one microservice, one user story with one estimated story point. Then Cg1=0, Max(P1)=1, N=1, WsicT=1, Pf1=0, SIY=0, and Cx=2. Therefore Cx0=2.

• Development Time— (Ti): estimated development time (in hours) for microservice i, calculated by adding the estimated time of each user story in it. The longest development time is used to compare the decompositions.

• Granularity— (Gm): indicator of how good or bad the system decomposition is, according to its coupling (CpT), cohesion (CohT), number of user stories associated with the microservice (WsicT), points of cognitive complexity (CxT), and semantic similarity (SsT); it is calculated as the norm of the vector with these metrics MT→=[CpT,CohT,CxT,WsicT,(100−SsT)].

(14) Gm=|MT→|=CpT2+CohT2+CxT2+WsicT2+(100−SsT)2

Microservices are developed around business functions, and ideally each microservice is managed by a separate team. For this evaluation, we have assumed that each microservice is developed independently at the same time, so that the estimated development time (T) of the system is the longest estimated development time of the microservices in the application. This is a simplification, as in real life a development team may develop multiple microservices, and multiple microservices may be developed sequentially; this limitation will be considered in future work (Vera-Rivera et al., 2021).

For further detailed explanation and formalization of these metrics, see Vera-Rivera et al. (2021).

Objective function

In the proposed genetic algorithm of the microservices backlog (Vera-Rivera et al., 2021), the adaptation function combines the metrics of coupling (CpT), cohesion (CohT), granularity (WsicT), complexity (CxT), and semantic similarity (SsT). We test several objective functions (see Eq. (15)). Among the objective functions (F1 to F8) of the genetic algorithm, we select the best result for comparison with Semgromi.

(15) F1=(10CpT)2+CxT2+WsicT2+(100−SsT)2F2=(10CpT)2+WsicT2+(100−SsT)2F3=CxT2+(100−SsT)2F4=(10CpT)2+CohT2+(100−SsT)2F5=(10CpT)2+(100−SsT)2F6=(10CpT)2+CohT2+WsicT2+(100−SsT)2F7=(10CpT)2+CxT2+(100−SsT)2F8=(10CpT)2+CohT2+WsicT2

The genetic algorithm seeks to find the best combination, the best allocation of user stories to microservices in such a way that the objective function is smaller; it is iterative, in each iteration the best individuals are selected, each one has a chromosome, which is crossed with another individual to generate the new population (reproduction), some mutations are generated to find the optimal solution to the problem. In genetic selection processes, the strongest survive; in the case of the problem of automatic generation of the assignment of user stories to microservices, the n individuals that best fit the conditions of the problem survive; they correspond to the assignments that involve a smaller objective function. The objective functions were used in the evaluation projects (Cargo Tracking, JPet Store, Foristom Conferences, and Sinplafut), we selected the best results and compared it with Semgromi and the state-of-the-art methods.

Evaluation results

The detailed results of using the proposed technique on the project dataset are shown in Table 3. Figure 3 summarizes coupling (CpT), lack of cohesion (CohT), microservice weight (WsicT), and calls among microservices.

Table 3 Results for evaluation projects

Project	Method	N	Gm	CpT	CohT	SsT	W	CxT	T	MP	Cl	AC	
Cargo	Genetic	3	85.8	3.16	1.16	70.9	6	74.0	35	23	3	1.0	
Tracking	Semgromi	4	185.2	4.69	1.50	88.4	9	178.5	54	35	8	2.0	
	DDD	4	156.6	5.29	1.50	74.1	6	145.0	39	27	9	2.3	
	Sevice Cutter	3	206.8	3.16	1.15	74.4	10	202.5	61	41	8	2.7	
	MITIA	4	203.1	6.78	1.06	76.7	5	190.0	30	19	12	3.0	
Jpet Store	Genetic	5	104.7	1.41	1.79	86.5	9	102.5	54	35	3	0.6	
	Semgromi	5	143.6	2.83	1.79	86.6	6	140.5	32	20	7	1.4	
	DDD	4	203.7	3.46	1.50	85.3	8	200.0	36	22	9	2.3	
	Execution Traces	4	179.7	3.46	1.50	84.1	7	175.5	31	19	8	2.0	
Foristom	Genetic	4	56.3	0.00	1.50	74.4	8	49.5	134	67	0	0	
	Semgromi	5	470.0	4.90	1.79	73.9	13	466.5	176	90	7	1.4	
	DDD	4	428.0	3.16	1.50	75.7	9	426.0	167	83	6	1.5	
Sinplafut	Genetic	13	792.1	7.35	3.33	86.6	13	788.5	98	49	24	1.8	
	Semgromi	11	819.9	9.59	3.02	86.9	16	814.0	116	58	24	2.2	
	DDD	9	926.9	10.58	2.31	84.4	19	920.5	150	75	23	2.6	
	Architec	5	723.0	3.74	1.79	82.9	34	721.0	254	127	9	1.8	
Note:

Genetic: genetic algorithm of microservice backlog (Vera-Rivera et al., 2021).

Semgromi: semantic grouping algorithm of microservice backlog.

DDD: domain-driven design.

Service cutter: (Gysel et al., 2016).

MITIA: (Baresi, Garriga & De Renzis, 2017).

Execution traces: (Jin et al., 2019).

Architec: solution proposed by the architect or development team.

SsT value measured in percentage (%).

W: WsicT.

Cl: calls.

AC: average calls.

MP: Max. Pi.

T value measured in hours.

Figure 3 Graphical results of the metrics.

We analyze results for each metric below. Number of microservices in the system: All the methods analysed converged on almost the same number: CargoTracking three or four microservices, JPetStore to four or five microservices, Foristom Conferences to four or five microservices, and Sinplafut to nine or 11 microservices. Semgromi algorithm in some projects required microservice join operations to converge to the expected number of microservices, as did the genetic algorithm.

Coupling (CpT): had the lowest value for the genetic algorithm in all four projects, while Semgromi algorithm was lower than DDD and state of the art in only two projects.

Lack of Cohesion Metric (CohT): Both the genetic and Semgromi algorithms had similar but higher values than DDD and state-of-the-art methods.

Highest number of stories associated with a microservice (WsicT): In three out of four projects, the genetic algorithm had a lower value than DDD and the state-of-the-art methods, but Semgromi algorithm in two out of four projects the values are very close.

Calls between microservices: An indicator of application performance: the more calls between them, the more performance is affected, as algorithms have to be executed in different containers or machines, resulting in higher latency and longer execution times. In three of the four projects, the genetic algorithm gave a lower value, while in two of the four projects, Semgromi algorithm showed very small differences compared to DDD and state-of-the-art methods.

Cognitive complexity (CxT): it was significantly lower (Fig. 4) for the distribution obtained by the genetic algorithm in the four projects, whereas the Semgromi algorithm was lower than DDD and the state-of-the-art methods in only two projects (JPetStore and Sinplafut), and in the other two projects had similar complexity.

Granularity ( Gm): had results (Fig. 5) very similar to those obtained in complexity, with the genetic algorithm obtaining less complexity than DDD and the state-of-the-art methods; similarly, the Semgromi algorithm yield leses Gm in some projects (JPetStore and SSinplafut) and had a closely similar result in the others.

Figure 4 Cognitive complexity analysis.

Figure 5 Granularity metric analysis.

The global comparative analysis (Fig. 6) summarizes the results and comparisons between the genetic algorithm, the Semgromi algorithm and DDD. In the analyzed metrics, we conclude that the results obtained by the genetic algorithm and the Semgromi algorithm of Microservice Backlog are good alternatives to determine the granularity of the system’s microservices, with comparable or better results than the current state-of-the-art techniques.

Figure 6 Comparative analysis of the results.

Discussion

The SEMGROMI results are coherent from a functional and architectural point of view. The identified microservices can be implemented in this way. The candidate microservices have low coupling, low complexity, high cohesion and high semantic similarity.

The results show that SEMGROMI gives very similar or better results than DDD: similar number of microservices, lower coupling, lower lack of cohesion, lower number of stories associated with a microservice (WsicT), lower granularity metric (Gm), lower complexity by having fewer points associated with the microservice, and lower estimated development time. The development time is the longest of all the microservices. Since improvements in development time imply savings in project cost, a better decomposition leads to a reduction in project cost and computational resource usage.

These results are promising because DDD is the most widely used method for identifying microservices and their granularity. DDD is a manual method, where steps and procedures have to be followed to obtain the candidate microservices. We have proposed two semi-automatic algorithms, with little user involvement, that allow a faster, metric-driven definition of microservices. The decomposition is obtained automatically and is not based on the architect’s experience; although valuable, the user can participate, adapt, modify and compare the decomposition obtained by the method, improving the system using the proposed metrics.

According to the empirical evaluation, the lowest coupling was obtained with the Genetic Algorithm and then with the Semgromi Algorithm; the highest cohesion was obtained with the Genetic Algorithm and DDD. The grouping algorithm (SEMGROMI) shows close values, especially for cohesion. The lowest number of user stories assigned to a microservice was obtained by the genetic algorithm. The lowest granularity was obtained by the genetic algorithm, and the grouping algorithm (SEMGROMI) and DDD obtained similar results.

The lowest number of calls corresponds to our previous genetic algorithm, followed by DDD, which is close to our new grouping algorithm (SEMGROMI). This metric represents the degree of dependency of a microservice as part of an application; a higher value implies more latency (which affects performance) and more dependencies, as calls require the execution of operations in other microservices.

These metrics are estimates; they were calculated at design time from information contained in user stories and their dependencies. We look for microservices to be autonomous and independent, that they work by themselves, in a real system microservices need others to work, increasing the coupling, the calls define the number of times that one microservice must use another, while the request define how many times other microservices use the microservice, SEMGROMI allows us to analyze this metrics at design time.

Future work will need to validate them and determine how accurate they are with an application already in production. This validation is beyond the scope of this work.

To get better results than DDD, the joining and unjoining operations were fundamental; although in some cases similar or better results than DDD are obtained, we should always check that the user stories are associated in the right place; for example, the semantic similarity algorithm assumes that the training session is semantically very similar to the user session, but they are two different things. User review is important to analyze and evaluate the automatic results, suggest improvements, and get better results.

The scope of this work was defined the functional requirements specified as user stories following agile methodologies, these requirements detail the operations or services that the microservices must implement; the non-functional requirements, that may affect the proposed solution, were not considered. In agile methodologies the functional requirements can be managed as constraints of the user stories; these points will be considered as future work.

An important difference between the genetic algorithm and SEMGROMI in the Sinplafut case study was the execution time: the problem is complex, since increasing the number of user stories significantly increases the execution time. The average genetic algorithm took about 11.5 h in Sinplafut compared to the other case studies where the average execution time was 9.7 min; while SEMGROMI took about 1.5 min in Sinplafut and almost instantaneous in the other case studies. It was possible to identify that the calculation of metrics and semantic analysis are the ones that represent the highest computational cost of Microservices Backlog. Future work will address the computational cost with parallel computing, which was used and tested to generate the population of the genetic algorithm, it is intended to parallelize the calculation of the metrics and the calculation of the semantic similarity. The tests were performed using a core-i7 computer, with 16 gigabytes of Ram.

Conclusions

This study proposes an algorithm for semi-automatically grouping user stories into microservices, which is part of the Microservices Backlog model. It is compared to other methods for identifying microservices, such as domain-driven design, service cutter, microservices identification through interface analysis (MITIA), and our genetic programming method proposed in previous work.

As part of this study, three contributions are presented: (i) an algorithm to identify and evaluate the granularity of microservices, including the establishment of user stories associated with a microservice and thus the number of microservices associated with the application, (ii) identification and adaptation of a set of metrics to measure the complexity of microservices according to their coupling, cohesion, and size; and (iii) a mathematical formalization of a microservices-based application in terms of user stories and metrics.

The grouping algorithm (SEMGROMI) assigns n user stories to k microservices, grouping stories with the highest semantic similarity, i.e., grouping stories that refer to the same topic. It also groups the microservices that have the highest degree of coupling. The value of k is not fixed.

These results are promising because our method achieves better results than the other methods in many characteristics, especially with DDD, which is the most widely used method to identify microservices and their granularity. DDD is a manual method, and our proposed algorithm is an automatic method with little user involvement. After more rigorous validation, our model can become a useful and valuable tool for developers and architects.

The next steps of this research include: (i) validating the proposed algorithms in a real-world case study (work in progress), (ii) building the database of software projects with their user stories and microservices and validating the model with them, and (iii) validating the microservices backlog model in expert judgment.

Supplemental Information

Supplemental Information 1 User stories Foristom Conferences - Spanish.

Click here for additional data file.

Supplemental Information 2 User stories Jpet Store - Spanish.

Click here for additional data file.

Supplemental Information 3 User stories Sinplafut - Spanish.

Click here for additional data file.

Supplemental Information 4 User Stories Cargo Tracking.

Click here for additional data file.

Additional Information and Declarations

Competing Interests

Author Contributions

Data Availability

The authors declare that they have no competing interests.

Fredy H. Vera-Rivera conceived and designed the experiments, performed the experiments, analyzed the data, performed the computation work, prepared figures and/or tables, authored or reviewed drafts of the article, and approved the final draft.

Eduard Gilberto Puerto Cuadros conceived and designed the experiments, performed the experiments, analyzed the data, performed the computation work, authored or reviewed drafts of the article, and approved the final draft.

Boris Perez conceived and designed the experiments, performed the experiments, analyzed the data, prepared figures and/or tables, authored or reviewed drafts of the article, and approved the final draft.

Hernán Astudillo conceived and designed the experiments, performed the experiments, analyzed the data, performed the computation work, prepared figures and/or tables, authored or reviewed drafts of the article, and approved the final draft.

Carlos Gaona conceived and designed the experiments, performed the experiments, analyzed the data, authored or reviewed drafts of the article, and approved the final draft.

The following information was supplied regarding data availability:

The datasets are available in the Supplemental Files and the code is available at BitBucket: https://bitbucket.org/freve9/microservicesbacklog/.

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
