# Peer review of "SEMGROMI—a semantic grouping algorithm to identifying microservices using semantic similarity of user stories"

_PeerJ Computer Science, doi:10.7717/peerj-cs.1380_

## Round 0.1 · original submission · Major Revisions

Please carefully address reviewer 1's main concerns about the experimental design. Also, please provide a replication package.

Then address all the other remarks focusing on the writing style and Figure quality.

Reviewer 1 ·

Basic reporting

The authors proposed a semi-automatic approach for identifying candidate microservices for designing an MSA based applications. The proposed approach makes use of user stories; it analyzes them, group them based on their semantic similarities and use a set of metrics (cohesion and coupling) to identify the quality of produced design.

The approach is interesting and has been validated with state-of-the-art techniques and methods regarding four case studies.

The language of the manuscript is understandable, but the writing style could be improved.

The authors cited a set of recent and relevant literatures. However, some of them are cited with missing details in the reference section: lines 511, 513, 562 (add a link to the library).

For self-containment, the authors should include the details of formalizations of used metrics at line 370 instead of redirecting readers to check the authors' previous paper.

Experimental design

Here are concerns that need to be considered to clarify the proposed approach:

1. Is there any template used to describe user stories? Templates may help to extract relevant information in more systematic way and similar entities from full descriptions.

2. What is and entity in your system, are the single words or did you use n-grams? in the case of using n-grams, please explicitly state the n value.

3. The role of c parameter in equation 2 is not clear. Please clarify

4. The authors stated in page 7 that: “For each pair MSi and MSj, the algorithm calculates the number of times that MSi calls MSj (callsi) and viceversa, (requesti). If both values are greater than zero, then microservice MSi and MSj are said to be interdependent, and therefore must be joined into one.”, if we adopt such rule as it is, the system will end up with a single microservice. Some interdependencies (a threshold value) should be accepted in real world systems !

5. In Algorithm 1, the authors should clearly specify how calls and requests are calculated from texts (list of frequent entities). Later, the authors added “User story dependencies were identified according to the data flow, calls, invocations among user stories, or business logic” and added in the discussion section that “These metrics are estimates; they were calculated at design time from information contained in user stories and their dependencies”. How exactly these metrics could be measured?

6. One of the well-known limitations of adopting user-stories to design complex and real-world systems is their unsuitability to identify non-functional requirements compared with the DDD method. This needs to be discussed in the discussion section.

7. Another point to discuss is the reusability of available in-house or external microservices. By using such artifacts, the time of development process can considerably be reduced despite of the coupling and cohesion values. The proposed system only considers developing MS-based systems from scratch!?

Validity of the findings

Conclusions are consistent with the presented results. But no replication is possible due to the absence of details given for the proposed approach (see the section above).

·

Basic reporting

This paper proposes an algorithm, named SEMGROMI, for identifying candidate microservices considering the semantic similarity of user stories.
Some strong and weak points of the paper:
Strong points:
- The language is clear with professional English with minor typos and syntactical errors (detailed below).
- The paper is well-structured and its reading is smooth.
- The Introduction and background clearly identifies the problem and the novelty of the proposed approach.
- Methodology and implementation details are adequately analyzed.
- Validation and experimental results are very detailed and clearly highlight the innovation of the proposed approach.
- Conclusions are well stated and linked to the original research question,

Weak points (proposals for enhancement):
- The Introduction section lacks a final paragraph describing the structure of the rest of the paper.
- Some references should be placed at the first appearance of the related text (e.g. validation projects at lines 78-79)
- The related work section should provide more details. Especially the 2nd and 3rd paragraph of this section seem to be an enumeration of related works without giving essential details. In addition, a comparison table at the end of the section would be helpful.
- Objective functions of equation 5 are not clearly explained in the text.
- How the default value of 0.85 is initially provided? Is it arbitrary? The same of the coupling threshold (lines 198 and 203)
- Figure 3 is not clear. Maybe it could split into subfigure with higher resolution.
- Tables 2-5 are very detailed. Maybe a summarization of the validation results of these tables could be summed up to a single table or figure.

- Figures 5-6 should be placed earlier, before the references.
- Some typos and syntactical errors
*line 20. no dot after networks
*line 52. Merge this line with the next paragraph
*line 65. We have proposed in a previous work....
*line 94. A dependency....
*line 102. ...automatically.
*line 121 (DDD) 3Ds
*line 139. ... a three-step...
*line 253. vice versa
*line 374 (equation 5) in parenthesis
*line 467. An algorithm for identifying and evaluating...
*line 103. the best assignment....

Experimental design

- The paper clearly identifies the research problem, which is within the scope the the journal.
- The methodology used is sufficient and adequately detailed by the authors.
- Experiments clearly exemplifies the applicability of the approach.

Validity of the findings

- The novelty of the paper is clear, as it is directly compared with existing well established algorithms (e.g. DDD).
- The validation results are statistically sound and well-provided.

Reviewer 3 ·

Basic reporting

Some issues with English. E.g., Lines 97-98, " System through the metric calculator component calculates the metrics for analyzing the application." This should presumably be something like, "We use a system based on...to calculate..."

Another example: in Algorithm 1, Line 297, should "interdepent" be "interdependent"?

Some of the figures are fuzzy/too small. E.g., Figure 1 has fuzzy-looking text. The graphs in Figure 3 are indecipherable to this reader.

Experimental design

I think this is fine.

Validity of the findings

An issue I have is that the objective functions are not clearly identified up-front. What I mean is: the paper asserts, "The problem of assigning user stories to microservices is NP-hard," and cites Gao et al. 2020 as support. There are two issues I have with this. One is that it is unclear what, precisely, the problem of assigning user stories to microservices is. And why one should believe that it is NP-hard. And one cannot intuit this unless the problem of assigning user-stories to microservices is characterized sufficiently precisely.

Another issue is that while this paper is indeed correct that Gao et al. 2020 state, at least twice that I see, that *some* problem is NP-hard (not sure if it's the same problem as this paper), Gao et al.'s work provides no proof nor citation for their claims.

So this brings me to two points from my side. (1) I suggest that this paper remove this vacuous claim that the problem it studies is NP-hard...given that the problem is not characterized with sufficient precision to claim that, and, (2) This paper provide a precise characterization of the *problem* it addresses, e.g., via objective functions.

Another issue I have is that the material in the Section "Group user stories by semantic similarity" is stated quite abstractly. It really need some examples of user-stories, identifiers from those user stories and dictionary keys, for one to fully appreciate what's going on.

Additional comments

None.

---

## Round 0.2 · accepted · Accept

I am pleased to inform you that you have successfully addressed all of the reviewers' comments, and I am happy with the current version of the manuscript. I believe that your work is now ready for publication. Congratulations on this accomplishment!

·

Basic reporting

In terms of basic reporting, the revised manuscript adequately addresses the comments from the first review as well as those from the other reviewers, by providing clear and detailed information on the research question, methodology, and results. The manuscript now includes more precise and concise language, making it easier to understand. Furthermore, the authors have responded to each of the reviewers' comments in a systematic and transparent manner, demonstrating their commitment to improving the manuscript. In general, the revised manuscript meets the standards of basic reporting by providing sufficient and clear information that can be easily understood by readers.

Experimental design

Similarly, regarding experimental design, the authors have responded effectively to the comments provided in the initial review and from the other reviewers in the rebuttal letter. The revised manuscript now provides more details on the methodology and experimental procedures used.

Validity of the findings

In terms of the validity of the findings, the authors have made significant efforts to address the comments from the initial review and from the other reviewers in the rebuttal letter. The revised manuscript now provides a more detailed and convincing explanation of the results obtained, with a clear justification of their interpretation.